# Detailed Analysis of Neurological Symptoms and Sensory Disturbances Due to Chronic Arsenic Exposure in Toroku, Japan

**DOI:** 10.3390/ijerph182010749

**Published:** 2021-10-13

**Authors:** Takashi Sugiyama, Nobuyuki Ishii, Yuka Ebihara, Kazutaka Shiomi, Hitoshi Mochizuki

**Affiliations:** 1Division of Respirology, Rheumatology, Infectious Diseases and Neurology, Department of Internal Medicine, Faculty of Medicine, University of Miyazaki, Miyazaki 889-1692, Japan; shiomik@med.miyazaki-u.ac.jp (K.S.); mochizuk@med.miyazaki-u.ac.jp (H.M.); 2Department of Neurology, Chiyoda Hospital, Miyazaki 883-0064, Japan; nobuyuki_ishii@med.miyazaki-u.ac.jp; 3Department of Neurology, Ebihara General Hospital, Miyazaki 884-0006, Japan; yuka_iwakiri@med.miyazaki-u.ac.jp

**Keywords:** chronic arsenic intoxication, neurological symptom, sensory disturbance, natural history, medical interviews

## Abstract

As a result of population growth and the development of tube wells, humans’ exposure to arsenic has increased over the past few decades. The natural course of organ damage secondary to arsenic exposure is not yet well understood. In Toroku, Japan, an arsenic mine was intermittently operated from 1920 to 1962, and residents were exposed to high concentrations of arsenic. In this paper, we analyzed 190 consecutive residents for whom detailed records of neurological symptoms and findings were obtained from 1974 to 2005. All participants were interviewed regarding the presence of general, skin, hearing, respiratory, and neurological symptoms. Neurological symptoms were classified into extremity numbness or pain, constipation, dyshidrosis, sensory loss, and muscle atrophy. Superficial and vibratory sensation was also evaluated. More than 80% of participants experienced extremity numbness, and numbness was the most common neurological symptom. Numbness was associated with superficial sensory disturbance, and was correlated with the subsequent development of other neurological symptoms, including autonomic and motor symptoms. No previous studies have investigated the natural course of chronic arsenic intoxication; thus, these data serve as a guide for detecting early symptoms due to arsenic exposure.

## 1. Introduction

Heavy metals are harmful to human health and have long been abundant in the Earth’s crust [1]. In recent years, population growth and the development of tube wells have resulted in increasing human exposure to heavy metals [2]. In particular, arsenic has been used as a pesticide, a chemotherapeutic agent, and a constituent of consumer products. It can cause acute organ damage in the short term, as well as various health impairments, such as peripheral neuropathy, following chronic, low-concentration exposure [3,4]. Today, more than 150 million people worldwide are chronically exposed to arsenic [2]. However, the natural course of arsenic-related health problems is not yet well understood, even though this information could provide valuable knowledge regarding potential treatments.

In Toroku, a small village of 300–400 residents in Miyazaki Prefecture, Japan, an arsenic mine was intermittently operated from 1920 to 1962. Arsenic was actively roasted to produce arsenious acid from 1933 to 1941 and from 1955 to 1962 [5,6]. During these years, more than 300 residents were exposed to arsenic via food, water, air, and skin. Arsenic concentrations in the environment were first measured by Miyazaki Prefecture in 1972 [6], although these measurements were never performed during the mine’s operation between 1920 and 1962. According to the Miyazaki Prefecture report [6], many Toroku residents continued to be exposed to arsenic in contaminated drinking water after 1962, and the mean arsenic concentration in the hair of Toroku residents (N = 29) was 1.52 mg/kg in 1972 [5,6]. The arsenic concentration in the Toroku River has decreased over time, but even in 1975, it exceeded 50 ppb at most measurement points, and was partially used for drinking and rice field water [7]. Toroku residents could not use the water supply from other source and had to live a self-sufficient life with Toroku River, so they had no chance but to use polluted water. The arsenic exposure of Toroku residents seemed to have improved by 1980 [7], however, they were unable to avoid exposure to arsenic for decades, even after the mine was closed.

The Department of Neurology of the University of Miyazaki has conducted neurological examinations of Toroku residents since 1975. We previously reported the clinical features and electrophysiological findings of 137 participants who had lived in Toroku [8]. The longitudinal sequelae of arsenic intoxication were described in our descriptive study [8], but we were unable to investigate the natural course of neurological impairments caused by arsenic exposure. Accordingly, we analyzed 190 consecutive participants for whom detailed neurological symptoms and findings were recorded by 2002.

## 2. Materials and Methods

### 2.1. Study Design

This study used a retrospective cohort design. The study protocol was approved by the Ethics Committee of the University of Miyazaki, with a waiver of written informed consent from participants with chronic arsenic exposure, and was carried out in accordance with the Declaration of Helsinki [9].

### 2.2. Participants

We enrolled participants who had lived within 1000 m of the mine roaster prior to 1962. They had typical symptoms and signs of arsenic poisoning, such as dermatological manifestations (pigmentation or keratosis) and symmetric neurological disturbances (peripheral neuropathy), and were diagnosed as “chronically exposed to arsenic” by the Japanese government after the Toroku Medical Examination began in 1974. 

According to medical interviews and/or annual blood laboratory data, 12 patients developed diabetes and 3 underwent gastrectomy, all at age 65 or older. The effects of these diseases were considered small, and these 15 participants were included in the study. No patients had alcoholism.

### 2.3. Symptoms

All participants who underwent medical examinations from 1974 to 2005 were interviewed regarding the presence of various symptoms lasting 1 year or longer. Most symptoms identified during the initial interview were later confirmed to have persisted beyond that time. The symptoms comprised general, skin, hearing, respiratory, and neurological issues. Each of these general categories was classified into a number of detailed symptoms. General symptoms were classified into headache and dizziness; dermatological symptoms were classified into skin disturbance, teeth problems, and nail diseases; otolaryngological symptoms were classified into olfactory dysfunction, tinnitus, and deafness; respiratory symptoms were classified into sore throat, cough, and wheezing; and neurological symptoms were classified into extremity numbness or pain, constipation, dyshidrosis, sensory loss, and muscle atrophy.

### 2.4. Objective Sensory Disturbances

Superficial sensation was evaluated with a sharp toothpick for pain and with test tubes containing water at 0 °C or 50 °C for temperature, and vibratory sensation was evaluated using a 128-Hz tuning fork. Abnormal superficial sensation (pain, cold, and hot) was defined as a decrease of 70% or less or hypersensitivity of 110% or more in the distal extremities compared with the trunk. Abnormal vibratory sensation was defined as a decrease of 70% or less in the distal extremities compared with the trunk. If objective neurological findings demonstrated a clear left–right difference or a clear limb difference, the side with milder sensory disturbance was adopted to reduce the possibility that another condition, such as cerebral infarction or orthopedic disease, was contributing to the symptoms.

### 2.5. Duration between Initial Arsenic Exposure and the Onset of Each Symptom

The arsenic mine engaged in active roasting from 1933 to 1941, and high-concentration arsenic contaminated the drinking water for a while thereafter. Therefore, we defined initial exposure to high concentrations of arsenic as beginning in 1933. 

For each participant, we calculated the duration (years) from initial arsenic exposure to the onset of symptoms. The year of initial arsenic exposure was defined as 1933 if the participant was born before 1933, or as the year of birth if the participant was born in 1933 or later.

### 2.6. Statistical Analysis

First, the average duration from initial exposure to the onset of each symptom was compared. The mean duration of numbness, which was the symptom that usually occurred soonest (Figure 1), was compared with the mean duration of other neurological symptoms using the paired *t*-test in participants who had both numbness and another symptom. Second, Pearson’s chi-square test was used to determine whether numbness was correlated with objective superficial or vibratory sensory disturbances. Finally, sensitivity, specificity, and the McNemar test were used to determine whether preexisting numbness was associated with the presence of other subsequent neurological symptoms in participants who had both numbness and another symptom. We excluded participants in whom numbness and another neurological symptom occurred during the same year. The statistical significance level was set at *p* = 0.05. Statistical analyses were performed using EZR software version 1.30 [10].

## 3. Results

### 3.1. Participant Characteristics

The participant characteristics are shown in Table 1. Of 190 participants, 164 (86.3%) were born before 1932 and were exposed to high concentrations of arsenic after infancy, and the average age of initial arsenic exposure was 16.6 years. Twenty-six participants (13.7%) were born after 1933. The average age of participants when the mine closed in 1962 was 42.7 years. 

### 3.2. Natural History of Reported Symptoms

The number and percentage of participants with each symptom are summarized in Table 2. Headache, skin disturbance, and extremity numbness were reported by more than 80% of participants. The duration from initial arsenic exposure to the onset of each symptom is shown in Figure 1. More common symptoms appeared earlier than the other symptoms in the general, dermatology, and neurology categories. 

In particular, extremity numbness developed considerably earlier than any other neurological symptoms. However, we could not confirm that numbness was the first neurological symptom in patients with chronic arsenic exposure based only on the data in Figure 1. Differences in the duration from initial arsenic exposure to the onset of numbness and other neurological symptoms were analyzed with the paired *t*-test, as shown in Figure 2. We confirmed that extremity numbness was the earliest neurological symptom in participants who suffered from numbness and another symptom in this category.

### 3.3. Numbness and Objective Sensory Disturbances

We analyzed the relationship between numbness and objective sensory disturbances in the 177 participants for whom these data were recorded. The results are shown in Table 3. Most participants with objective superficial sensory disturbances experienced extremity numbness (104/111; 93.7%), and a Pearson’s chi-square analysis of independence indicated that the presence of numbness was significantly associated with impaired superficial sensation (*p* = 0.027). While most participants with objective vibratory sensory disturbance also had extremity numbness (118/140; 90.8%), the Pearson’s chi-square analysis of independence showed no significant relationship between the two symptoms (*p* = 0.492). 

### 3.4. The Relationship between Numbness and Other Neurological Symptoms

Figure 1 and Figure 2 show that on average, numbness was the earliest neurological symptom. In a further analysis, we determined the proportions of participants with numbness who subsequently developed other neurological symptoms, as shown in Figure 3. More than 90% of participants experienced sensory loss or muscle atrophy after feeling numbness. Similarly, 76.6% of participants developed dyshidrosis. According to these results, numbness was a highly sensitive marker for predicting the occurrence of other neurological symptoms (sensitivity > 0.750, *p* < 0.0001), but it had low specificity (<0.250).

## 4. Discussion

This study investigated the natural history of individuals with chronic arsenic exposure. Extremity numbness was experienced by more than 80% of participants, and was the usually first neurological symptom to occur. Preexisting numbness was related to superficial sensory disturbance and to the subsequent development of other neurological symptoms, including autonomic and motor symptoms. No previous studies have assessed the natural course of chronic arsenic intoxication; thus, this research should provide a foundation for detecting early symptoms due to arsenic exposure.

### 4.1. Extremity Numbness Is Often the First Symptom in the Natural History of Arsenic Exposure

More than 80% of participants experienced extremity numbness, and this condition was usually the first neurological symptom. In a study from West Bengal, India, common symptoms were paresthesia and muscle weakness [11]. A study conducted in Myanmar examined the symptoms resulting from long-term exposure to low concentrations of arsenic in drinking water, and identified pain and vibration sensory disturbances, but few motor symptoms [12]. 

The results of this study suggest that numbness, classified as a type of paresthesia, is often one of the earliest symptoms of arsenic exposure. Numbness predicted other neurological symptoms with a high sensitivity; therefore, it is important to screen for numbness in the interviews of patients exposed to arsenic. 

### 4.2. Numbness and Small Fiber Neuropathy Due to Arsenic Exposure

Pain and vibration sensations depend on the functions of small and large peripheral nerve fibers, respectively. Several studies showed that arsenic exposure induced peripheral neuropathy or neuritis, and that sensory nerves were more commonly impaired than motor nerves [4,11,13,14,15,16]. A few studies separately analyzed the functions of small and large fibers [5,8,12], and one suggested that based on clinical observations, the aforementioned effects of arsenic particularly affected small fibers [5]. Large-fiber function can be evaluated quantitatively by nerve conduction studies. A study performed in Inner Mongolia reported that chronic arsenic exposure from drinking water (mean arsenic concentration in tube well, 158.3 ppb) did not affect the nerve conduction velocity [17]. In the Myanmar study, the authors found that both small and large peripheral nerve fibers were impaired by drinking water containing low concentrations of arsenic [12]. In addition, it was reported that exposure to multiple heavy metals, including arsenic, caused significant damage to small-diameter fibers [18].

In this study, numbness was significantly associated with superficial sensory disturbance, but not with impairment in vibratory sensation. Consistent with a previous report [5], these results suggest that arsenic exposure damages small-diameter fibers at an early stage. 

### 4.3. Limitations

This study has two limitations. First, arsenic exposure was not constant over a long period of time. The results concerning the duration of exposure and time to symptom onset in this study are unlikely to be applicable to other arsenic-exposed areas. Second, participants were examined long after their last exposure to arsenic, and the participants were relatively old. Elderly individuals have multiple comorbidities besides diabetes, and many of them suffer from peripheral neuropathy. Therefore, it is difficult to rule out the possibility that other factors may have contributed to their peripheral neuropathy.

## 5. Conclusions

Individuals with chronic arsenic exposure reported numerous neurological symptoms, and more than 80% experienced extremity numbness. Numbness occurred earlier than the other neurological symptoms that impaired quality of life, including autonomic and motor symptoms. As numbness was identified through medical interviews, such interviews are as important as neurological examinations to detect the initial symptoms of chronic arsenic exposure.

## Figures and Tables

**Figure 1 ijerph-18-10749-f001:**
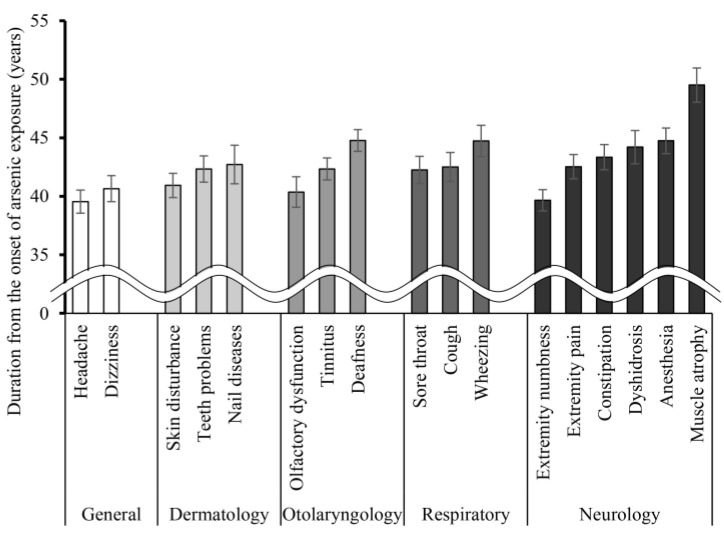
The duration from initial arsenic exposure to the onset of symptoms. In each category, symptoms are ordered according to the time between their onset and the initial arsenic exposure. Bar graphs are presented as mean ± SEM.

**Figure 2 ijerph-18-10749-f002:**
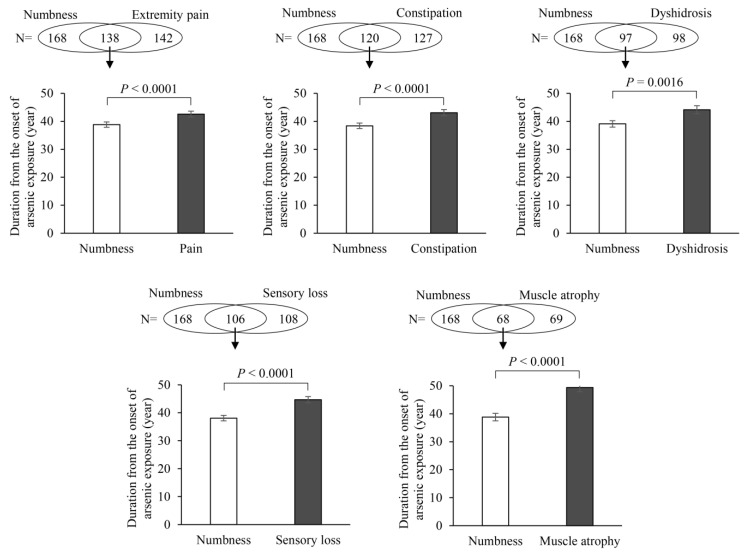
Differences in the duration from initial exposure to symptom onset between numbness and other neurological symptoms were analyzed with the paired *t*-test in participants with numbness and at least one other neurological symptom. We confirmed that extremity numbness was the earliest subjective neurological symptom. Bar graphs are presented as mean ± SEM.

**Figure 3 ijerph-18-10749-f003:**
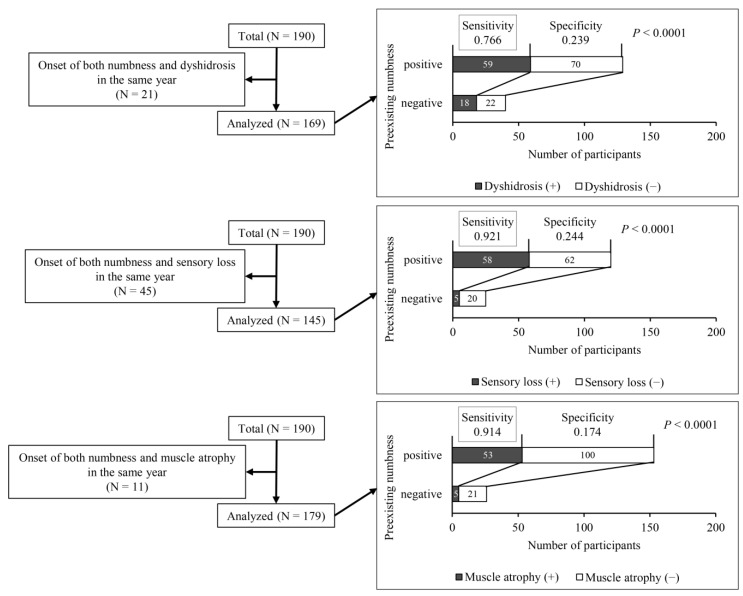
The relation between numbness and other subsequent neurological symptoms, specifically dyshidrosis, sensory loss, and muscle atrophy, in participants who had numbness and at least one other neurological symptom. Sensitivity and specificity were analyzed. Significant differences were analyzed by the McNemar test.

**Table 1 ijerph-18-10749-t001:** Participant characteristics.

Parameters	Total (N = 190)
Male/female, n	100/90
Participants with active arsenic exposure	
from birth	26 (13.7) ^(1)^
after infancy	164 (86.3) ^(1)^
Initial age of first arsenic exposure	16.6 (1–44) ^(2)^
Age in 1962 when the mine closed	42.7 (16–73) ^(2)^

N, number; ^(1)^, number (%); ^(2)^, average number of years (youngest–oldest).

**Table 2 ijerph-18-10749-t002:** Symptoms.

Parameters	Total (N = 190)
**General**	
Headache	155 (81.6)
Dizziness	144 (75.8)
**Dermatology**	
Skin disturbance	155 (81.6)
Teeth problems	128 (67.4)
Nail diseases	89 (46.8)
**Otolaryngology**	
Tinnitus	141 (74.2)
Deafness	134 (70.5)
Olfactory dysfunction	114 (60.0)
**Respiratory**	
Cough	128 (67.4)
Sore throat	107 (56.3)
Wheezing	106 (55.8)
**Neurology**	
Extremity numbness	168 (88.4)
Extremity pain	142 (74.7)
Constipation	127 (66.8)
Sensory loss	108 (56.8)
Dyshidrosis	98 (51.6)
Muscle atrophy	69 (36.3)

Categorical variables are shown as numbers (percentages).

**Table 3 ijerph-18-10749-t003:** Relationship between numbness and objective sensory disturbances.

		Numbness			Pearson’s Chi-Square Test	
		Positive	Negative	Value	df	*p*
Superficial sensory disturbance	positive	104	7			
	negative	55	11	4.863	1	0.027

Vibratory sensory disturbance	positive	118	12			
	negative	41	6	0.472	1	0.492

The numbers in the “Numbness” columns indicate the number of participants.

## Data Availability

The data presented in this study are available from the corresponding author upon reasonable request. The data are not publicly available due to participants’ privacy.

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
