# Peer review of "Detailed Analysis of Neurological Symptoms and Sensory Disturbances Due to Chronic Arsenic Exposure in Toroku, Japan"

_ijerph, 2021, doi:10.3390/ijerph182010749_

Round 1

Reviewer 1 Report

This is a well written paper, and includes some very important information about the health effects of arsenic, especially focused on neuropathy associated with arsenic.  This health effect is not often discussed in the literature, so hopefully this paper will spark the study for the mechanism of arsenic leading to this health effect.  It is also good to see this type of data being published to make readers and the public aware of the many health consequences of arsenic exposure.   The only minor criticism that I have is that it would have been good to know what was the exact contribution of arsenic from the diet and drinking water (and perhaps even airborne exposure to arsenic contaminated dust) in the area.

Author Response

We thank this reviewer for thoughtful comments.

The only minor criticism that I have is that it would have been good to know what was the exact contribution of arsenic from the diet and drinking water (and perhaps even airborne exposure to arsenic contaminated dust) in the area.

Reply)

Thank you for the helpful comment and sorry for our poor presentation.

We agree the reviewer's criticism, and it is difficult to estimate an exact arsenic intake. However, it is certain that Toroku residents had difficulty drinking arsenic-free water and had no choice but to consume contaminated water and food.

We have added and revised below sentences in 1. Introduction (Line 49-53).

“Toroku residents could not use water supply from other source and had to live a self-sufficient life with Toroku River, so they had no chance but to use polluted water. The arsenic exposure of Toroku residents seemed to have improved by 1980 [7], how-ever, they were unable to avoid exposure to arsenic for decades even after the mine was closed.”

Reviewer 2 Report

The paper entitled “Detailed Analysis of Neurological Symptoms and Sensory Disturbances Due to Chronic Arsenic Exposure in Toroku, Japan” is well structured and presented. The main topics of the paper are well introduced, and the methodology and results are well organized.

I suggest that the authors should reflect about the limitations of the study, namely the age of the participants and the possibility of other confounding variables affecting the study population, like other pathologies present in the participants.

Author Response

We thank this reviewer for thoughtful comments.

I suggest that the authors should reflect about the limitations of the study, namely the age of the participants and the possibility of other confounding variables affecting the study population, like other pathologies present in the participants.

Reply)

We agree the reviewer’s comment and add a below sentence in 4.3. Limitation (Line 226-227).

“Elderly individuals have multiple comorbidities besides diabetes, and many of them suffer from peripheral neuropathy.”

Reviewer 3 Report

The manuscript entitled ‘’Detailed analysis of neurological symptoms and sensory disturbances due to chronic arsenic exposure in Toroku, Japan’’ used a retrospective cohort design to analyze the natural course of neurological impairments related to chronic exposure to arsenic.

This manuscript is well written and easy to follow. I think it is worth publishing provided minor revision.

Minor comments:

Figure 1 and Figure 2: titles of x and y axis look a bit blurry. Maybe the quality of the text is a bit low?

Lines 49 -51: ‘’This situation in which Toroku residents took arsenic daily seemed to have improved around 1980 [7], but they were unable to avoid exposure  to arsenic for decades after closure of the mine. ‘’

Please, revise this sentence. The verb ‘’take’’ might sound as if the action was intentional.

Lines 213- 214: ‘’..these results show that arsenic exposure damages small-diameter fibers at an early stage.’’

Instead of ‘’show’’ maybe it would be better ‘’suggest’’.

Author Response

We thank this reviewer for thoughtful comments.

Figure 1 and Figure 2: titles of x and y axis look a bit blurry. Maybe the quality of the text is a bit low?

Reply)

We understand this reviewer’s comments, and revise Figure 1, 2, and 3.

Lines 49 -51: ‘’This situation in which Toroku residents took arsenic daily seemed to have improved around 1980 [7], but they were unable to avoid exposure to arsenic for decades after closure of the mine. ‘’

Please, revise this sentence. The verb ‘’take’’ might sound as if the action was intentional.

Reply)

We understand these comments, and revised this sentence to the below sentence (Line 51-53).

“The arsenic exposure of Toroku residents seemed to have improved by 1980 [7], however, they were unable to avoid exposure to arsenic for decades even after the mine was closed.”

Lines 213- 214: ‘’..these results show that arsenic exposure damages small-diameter fibers at an early stage.’’

Instead of ‘’show’’ maybe it would be better ‘’suggest’’.

Reply)

We understand these comments, and revise “show” to “suggest” (Line 219).